# Poling-Free Hydroxyapatite/Polylactide Nanogenerator with Improved Piezoelectricity for Energy Harvesting

**DOI:** 10.3390/mi13060889

**Published:** 2022-05-31

**Authors:** Wei Liu, Yunlai Shi, Zhijun Sun, Li Zhang

**Affiliations:** State Key Laboratory of Mechanics and Control of Mechanical Structures, Nanjing University of Aeronautics and Astronautics, Nanjing 210016, China; bx2001039@nuaa.edu.cn (W.L.); meezjsun@nuaa.edu.cn (Z.S.); zhangli240@uestc.edu.cn (L.Z.)

**Keywords:** polylactide, hydroxyapatite, nanogenerators, energy harvesting

## Abstract

Polylactide-based piezoelectric nanogenerators were designed and fabricated with improved piezoelectric performances by blending polylactide with hydroxyapatite. The addition of hydroxyapatite significantly improves the crystallinity of polylactide and helps to form hydrogen bonds, which further improved the piezoelectric output performance of these piezoelectric nanogenerators with over three times the open circuit voltage compared with that of pure-polylactide-based devices. Such excellent piezoelectricity of hydroxyapatite/polylactide-based nanogenerators give them great potential for energy harvesting fields.

## 1. Introduction

With the energy crisis and environmental pollution become more and more serious, people have demanded and explored renewable energy technology to support our society [1,2,3]. One possible approach is using solar energy, biofuel, water, and wind energy to replace the traditional petrochemical resources. Another possible way is to develop ambient energy harvesting devices to satisfy the requirements of microwatts for low-power electronics. Large amounts of ambient mechanical energy exist in our living surroundings, such as vibration, noise, human breath, walking, and hand waving. Although the development of mechanical energy is valuable, such kinds of mechanical energy were not fully used until the invention of piezoelectric nanogenerators (PENGs) [4,5,6,7].

PENGs can convert ambient mechanical energy to electricity based on piezoelectric materials. In 2008, Wang and his colleagues developed the first PENG using a ZNO nanowire array [8]. Since then, many researchers have developed different kinds of nanogenerators based on piezoelectric materials. Piezoelectric materials primarily include inorganic and organic materials. Generally, the inorganic materials, such as lead zirconate titanate, possess higher piezoelectric constants, but are rigid and easy to break. The organic materials, such as Poly (vinylidene fluoride) (PVDF) and its copolymer, have lower piezoelectric constants, but are flexible, allowing large deformation. With the rapid development of wearable and implantable devices, the requirements for flexibility and biocompatibility in PENGs have been raised [9,10]. Thus, organic piezoelectric materials have attracted attention in recent years.

Compared with the most used and mature organic piezoelectric materials, PVDF, polylactide (PLLA) possesses several advantages, such as high biocompatibility, no requirement of poling to achieve piezoelectric performances, high heat-resistant properties, and renewable origins [11,12,13]. However, its relatively low piezoelectric constant has seriously limited its application in PENGs. Researchers have made efforts to improve the piezoelectric constants. By using different fabricated methods, including electro-spun or cast coating, or forming different structures, such as cantilevers or films, several PLLA-based nanogenerators have been made [14,15,16,17,18]. However, the piezoelectric constant improvement of PLLA is still limited.

Blending with a functional filler is an easy and effective way to improve the performances of polymers. However, little work has used fillers to improve the piezoelectric performances of PLLA. Meanwhile, a filler should not decrease the biocompatibility of PLLA. Thus, in this work, the biocompatible filler hydroxyapatite (HA) was blended with PLLA to improve its piezoelectric performance. The addition of HA significantly improved the β-phase crystal crystallinity of PLLA and further increased the piezoelectric output of the HA/PLLA PENG.

## 2. Experiment

### 2.1. Preparation of HA/PLLA Composite Films

HA (Chengdu Organic Chemicals Co., Ltd., Chengdu, China) and PLLA (NatureWorks, 2003D) were dispersed in trichloromethane (CHCl_3_). The mixed solution was magnetic-stirred and scrap-coated onto a silver nanowire with polyethylene terephthalate (PET) film. Then, the wet film was placed in a vacuum oven to evaporate the solution and form a 10 μm-thick composite film.

The composite film was annealed at 140 °C for 30 min. The pure PLLA film and composite films with HA contents of 10% wt, 20% wt, and 30% wt were prepared and named PLLA, 10%-HA/PLLA, 20%-HA/PLLA, and 30%-HA/PLLA, respectively.

### 2.2. Preparation of HA/PLLA PENG

Copper electrodes were magnetron-sputtered onto the prepared HA/PLLA composite film with silver nanowire on the PET film. The copper electrodes and the silver nanowire on the PET film serve as top and bottom electrodes, respectively. Then, a plastic cover was attached as packaging. The preparation process of the HA/PLLA PENGs is shown in Figure 1.

### 2.3. Characterization and Measurements

The FTIR spectra of the composite films were obtained using a Bruker Tensor 27 spectrometer. The spectra were obtained in the range from 4000 cm^−1^ to 500 cm^−1^. The X-ray diffraction (XRD) measurements were performed on a D/Max2500 VB2t/PC X-ray diffractometer (Rigaku, Japan) for a 2θ range of 5–50°. The energy harvesting performance, including the output voltage and load resistance behavior, of PLLA and HA/PLLA PENG was assessed using a high-speed-acquired card (NI 9308), as shown in Figure 2.

## 3. Results and Discussion

The crystalline behavior of the HA/PLLA composite films were explored using XRD. In Figure 3a, diffraction peaks of PLLA were found at 2θ = 17.1°and 19.3°, corresponding to the (200) and (203) planes of the α-phase crystal, respectively. Meanwhile, with the addition of HA, the intensity of these two peaks became weak, and a new diffraction peak was found at 2θ = 31.2° for the HA/PLLA composite films, which corresponded to the (003) plane of the β-phase crystal. With the content of HA increasing to 30 wt%, the intensity of the diffraction peaks of the α-phase crystal continued to decrease, while the intensity of the diffraction peaks of the β-phase crystal increased. Such phenomena indicated that the addition of HA to PLLA may help to enhance the transformation of β-phase crystal to α-phase crystal [19,20]. It is well known that the piezoelectricity of PLLA originates from the orientation of C=O dipoles. Since the β-phase crystal in PLLA often forms when the PLLA backbones are in a parallel state, the C=O dipoles are in a high-orientation state. Thus, the higher content of β-phase crystals in PLLA may help to enhance the output performances of the HA/PLLA PENG.

FTIR was used to investigate the interactions between HA and PLLA. In Figure 3b, the characteristic peak at 2875 cm^−1^ and 2985 cm^−1^ is due to the symmetric and asymmetric stretching of methylene groups in PLLA. The characteristic peak around 1700 cm^−1^ was attributed to the carbonyl groups in the ester bonds of PLLA. In addition, two new characteristic peaks were found around at 3400 cm^−1^ and 960 cm^−1^, respectively, due to the addition of HA to PLLA. Among them, the characteristic peak at 955 cm^−1^ belongs to the β-phase crystal in PLLA. The intensity of such a characteristic peak increased with higher HA content, implying that the β-phase crystal content increased with increasing HA content. The characteristic peak at 3400 cm^−1^ was attributed to hydrogen bonds between carbonyl in PLLA and the hydroxy in HA. Meanwhile, with higher HA content, the characteristic peak at 3400 cm^−1^ gradually moved to a lower wave number area, which confirmed the formation of stronger hydrogen bonds. Such interfacial interactions between PLLA and HA may help to improve the piezoelectric output of the HA/PLLA PENG.

The interfacial interactions and cross-section morphology of the HA/PLLA composites were further investigated by SEM. In Figure 4, the light phase is the HA filler and the dark phase is the PLLA matrix. The SEM images with different magnifications show that the HA dispersed uniformly in the PLLA matrix. At higher magnification, we can see that the HA bonded well with the PLLA matrix with indistinct edges, indicating interfacial interactions between HA and PLLA, which is accordance with the FTIR analysis.

The output performances of the HA/PLLA PENGs are provided in Figure 5. Under the same impact conditions, the PLLA PENG output only 1 mV voltage. With the addition of HA to PLLA, the output voltage improved significantly, reaching the maximum voltage of ~3.4 mV when the HA content was 20%, which was over three times that of the pure-PLLA PENG. According to the XRD and FTIR analysis, the addition of HA to PLLA may help to form β-phase crystals and hydrogen bonds between PLLA and HA, which may improve the piezoelectric output of the HA/PLLA PENGs. However, with further increase in HA content, the output voltage of HA/PLLA PENG decreased to 0.7 mV, which was possibly due to the aggregation of the HA filler in the PLLA matrix.

As the HA-20/PLLA showed the optimum piezoelectric output, the impedance properties of HA-20/PLLA PENG were further investigated. With the increase in load resistance from 0 to 120 MΩ, the output voltage increased from 0 to 4 V. Meanwhile, the output current decreased from 0.35 mA to 0.04 mA. According to the test results, the maximum power of HA-20/PLLA PENG was 5 μW when the load resistance was 1.3 MΩ. The durability of HA-20/PLLA PENG is provided in Figure 5. After 5000 impact cycles, the output voltage of HA-20/PLLA PENG remained constant, implying its good durability.

As numerous types of mechanical energy exist in our daily life, such as human walking or finger beating, the conversion of such mechanical energy to electric power using PENG is meaningful. To verify the practical application of HA/PLLA PENGs, we study the output performance of HA/PLLA PENGs under human motion conditions. In Figure 6b, the stress–strain curves of HA/PLLA PENG are provided, indicating its brittle fracture behavior and poor strain. The HA/PLLA PENG can output about 0.2 V and 6 V voltage under beating and treading conditions, respectively, which are higher values than those of the PLLA PENG. Such output voltage can supply some low-power electric devices.

## 4. Conclusions

In this paper, a novel PENG was prepared based on HA/PLLA composite films. The addition of HA to PLLA significantly improves the crystallinity of PLLA, and helps to form hydrogen bonds between PLLA and HA, both of which may increase the piezoelectric output of HA/PLLA PENGs. The HA/PLLA PENG with 20 wt% HA showed over three times the output voltage of neat PLLA PENG. The practical energy harvesting application of HA/PLLA for low-power electric devices was confirmed by finger beating and foot treading conditions.

## Figures and Tables

**Figure 1 micromachines-13-00889-f001:**
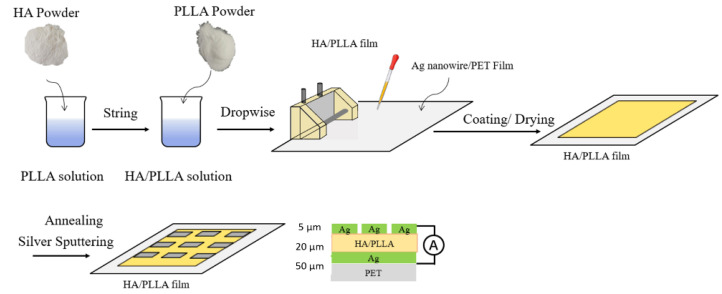
Optical images of rGO/PVDF-TrFE WNGs.

**Figure 2 micromachines-13-00889-f002:**
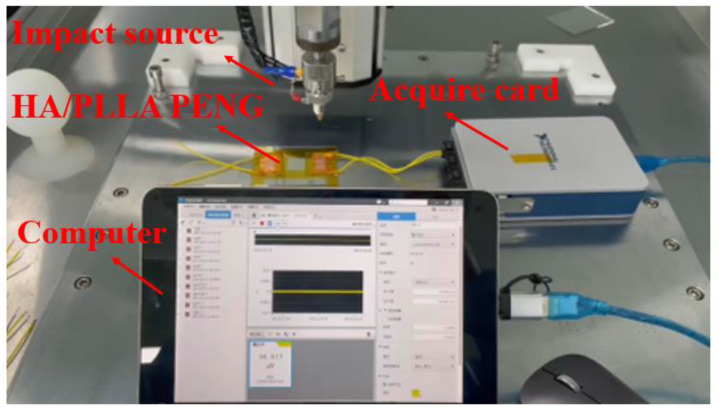
The evaluating system of PENG output performances.

**Figure 3 micromachines-13-00889-f003:**
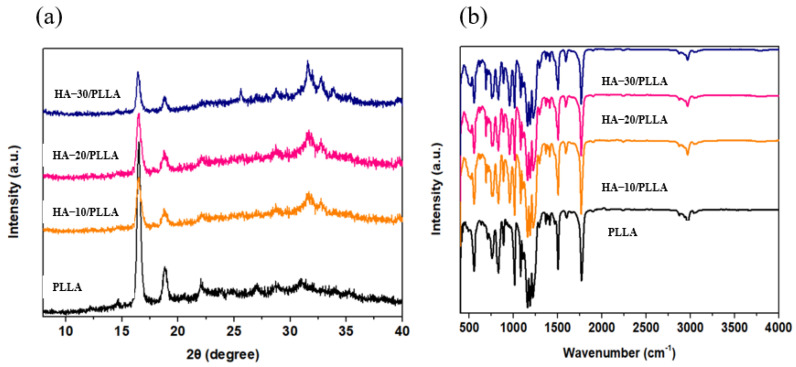
The (**a**) XRD and (**b**) FTIR spectra of PLLA and HA/PLLA composite films.

**Figure 4 micromachines-13-00889-f004:**
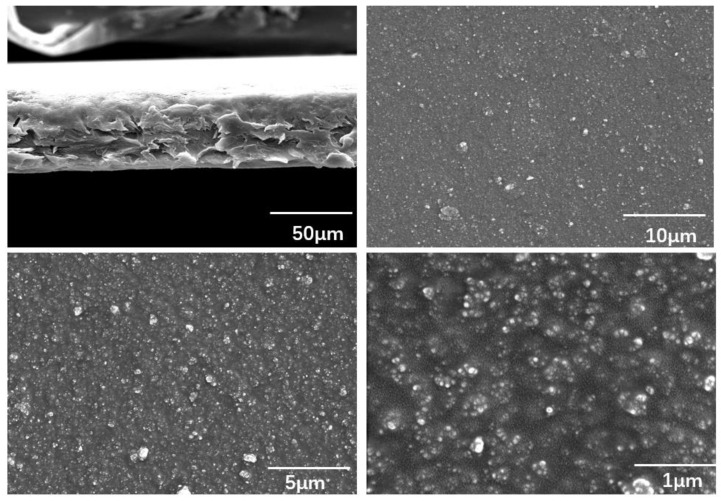
Cross-section SEM images of HA/PLLA composite film (20 wt%).

**Figure 5 micromachines-13-00889-f005:**
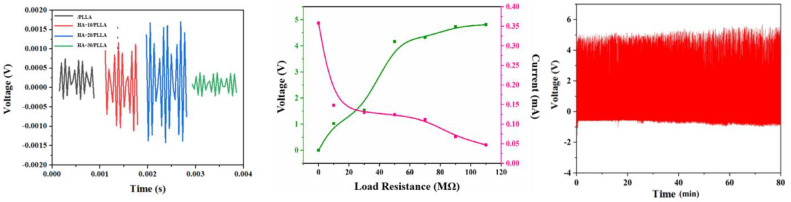
The output performances and durability of PLLA and HA/PLLA PENG.

**Figure 6 micromachines-13-00889-f006:**
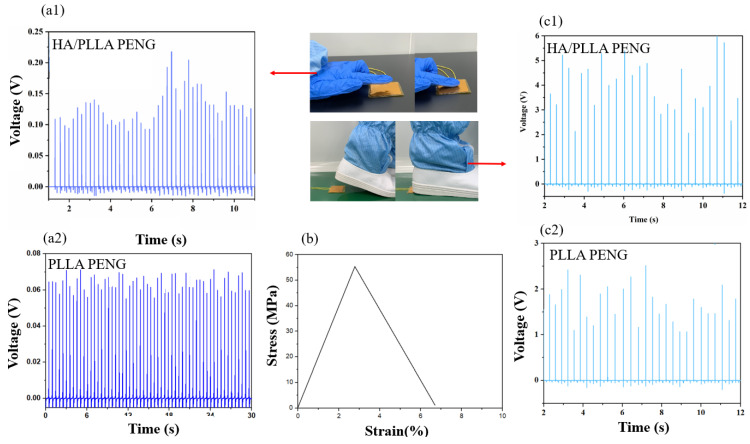
(**a1**,**a2**,**c1**,**c2**) Output performances of PLLA and HA/PLLA PENG under beating and treading conditions; (**b**) stress–strain curves of HA/PLLA PENG.

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
