# Peer review of "Poling-Free Hydroxyapatite/Polylactide Nanogenerator with Improved Piezoelectricity for Energy Harvesting"

_micromachines, 2022, doi:10.3390/mi13060889_

Round 1

Reviewer 1 Report

  1. Please match the PENG in the abstract part with the actual name. It would be more appropriate to delete all abbreviations.
  2. The abstract and conclusion should be rewritten.
  3. Please double-check the format of the refs..

  4. Please measure the stress-strain curve of the PENG in figure 6.
  5. Please add the durability test of the PENG in figure 5.

  6. Please give the device model of output voltage measurement in part 2.3.

  7. Please show cross-section images of composite film and the PENG in figure 4.

Reviewer 2 Report

Dear Authors,

My comments and questions are listed in the attachment.

Kind Regards

Round 2

Reviewer 2 Report

Dear Authors,

The manuscript can be published after minor revision. 

Kind Regards